# Gender-Specific Association of Serum Uric Acid and Pulmonary Function: Data from the Korea National Health and Nutrition Examination Survey

**DOI:** 10.3390/medicina57090953

**Published:** 2021-09-10

**Authors:** Hyemin Jeong, Sun-Young Baek, Seon-Woo Kim, Eun-Jung Park, Hyungjin Kim, Jaejoon Lee, Chan-Hong Jeon

**Affiliations:** 1Department of Internal Medicine, Division of Rheumatology, Soonchunhyang University Bucheon Hospital, Soonchunhyang University College of Medicine, 170, Jomaru-ro, Bucheon 14584, Korea; fly707@gmail.com; 2Biostatic and Clinical Epidemiology Center, Samsung Medical Center, 81 Irwon-Ro Gangnam-gu, Seoul 06351, Korea; sun0.baek@sbri.co.kr (S.-Y.B.); seonwoo.kim@samsung.com (S.-W.K.); 3National Medical Center, Department of Internal Medicine, Division of Rheumatology, 245, Eulji-ro, Jung-gu, Seoul 04564, Korea; ejcoke.park@gmail.com; 4Samsung Medical Center, Department of Internal Medicine, Division of Rheumatology, Sungkyunkwan University School of Medicine, 81 Irwon-Ro Gangnam-gu, Seoul 06351, Korea; chandler.kim@samsung.com

**Keywords:** uric acid, nutrition surveys, lung, respiratory function tests

## Abstract

*Background and Objectives*: Hyperuricemia is associated with several comorbidities. The association between uric acid (UA) and pulmonary function is still a controversial issue. This study evaluated the gender-specific association of serum UA and pulmonary function. *Materials and Methods*: A total of 3177 (weighted *n* = 19,770,902) participants aged 40 years or older were selected from the 2016 Korean National Health and Nutrition Examination Survey and included. *Results*: Female participants with hyperuricemia were older than participants with normouricemia. Body mass index (BMI), mean arterial pressure (MAP), hemoglobin A1c (HbA1c), and estimated glomerular filtration rate (eGFR) were significantly associated with UA levels in both males and females. Hyperuricemia and increase in UA quartile were significantly associated with decreased forced expiratory volume in 1 second (FEV1) and forced vital capacity (FVC) in females after adjustment for age, income, region, education, marital status, alcohol consumption, smoking, BMI, MAP, HbA1c, and eGFR. There was no significant association between UA levels and lung function in males. After additional adjustment for respiratory disease including pulmonary tuberculosis, asthma, and lung cancer, the association between hyperuricemia and decreased FEV1 and FVC in females was revealed. *Conclusions*: Hyperuricemia was associated with decreased FVE1 and FVC in the female general population.

## 1. Introduction

Serum uric acid (UA) is the end product of the metabolism of purine compounds in humans and higher primates. The UA level is determined by several factors including genetics, diet, alcohol, renal function, and drugs. Hyperuricemia is caused by overproduction of UA or underexcretion of UA or a combination of both processes. UA homeostasis is maintained in the human body by the kidneys and intestinal excretion. Decreased efficiency of renal UA excretion in patients is responsible for about 85% to 90% of cases of hyperuricemia. Hyperuricemia is associated with an increased risk of gout. Furthermore, hyperuricemia is related to hypertension, coronary heart disease, stroke, diabetes, obesity, and metabolic syndrome [1,2,3,4,5]. The mechanistic role of UA in several comorbidities suggested that UA stimulates vascular smooth muscle proliferation and oxidative stress [6]. Hyperuricemia is associated with endothelial dysfunction by inducing inhibition of nitric oxide production [7]. Hyperuricemia is also associated with the severity and mortality of chronic obstructive pulmonary disease (COPD) [8]. Subjects with lung restriction or airflow limitation had higher serum UA levels than subjects without lung restriction or airflow limitation in the general population in Japan [9], thus suggesting that oxidative stress and inflammation caused by UA induces lung tissue damage. On the contrary, Song et al. [10] reported that serum UA was positively associated with pulmonary function in the healthy population and suggested that hyperuricemia may have a positive effect on pulmonary function. A population-based cohort study in the United Kingdom showed that a lower level of UA was associated with higher rates of COPD and lung cancer in current smokers [11]. UA is an endogenous antioxidant and provides an antioxidant defense in humans against oxidant- and radical-caused diseases [12]. In the bronchial region, the UA level was reported to rapidly decrease after exposure to radical stress [13], thus suggesting that UA might provide an important first-line defense against oxidative stress such as smoking or air pollution. The outcomes of the published studies suggest that the association between UA and pulmonary function is still controversial. Therefore, the aim of this study was to evaluate the relationship between serum UA and pulmonary function in the Korean adult population using nationwide data.

## 2. Materials and Methods

### 2.1. Study Population

The Korea National Health and Nutrition Examination Survey (KNHANES) is a nationwide survey conducted periodically by the Korea Centers for Disease Control and Prevention to investigate health and nutritional status of the Korean population [14]. The KNHANES assesses the general health and nutrition status of individuals in South Korea through interviews about health and nutrition and basic health assessments. Participants were selected using the proportional allocation, systematic sampling method with multistage stratification to derive a representative Korean population. Although individual participants are not generally representative of the Korean population, this survey provides representative estimates of the non-institutionalized Korean civilian population by using the power of sample weight [15]. Every year, 10,000 to 12,000 individuals in approximately 3800 to 4600 households are selected from a panel based on the national census data. In the past several cycles of the KNHANES, participation rates of selected households has been high, ranging from 75% to 80%. The current study analyzed data from the 2016 KNHANES. In the 2016 KNHANES, 10,806 individuals in 4416 households were selected. The participation rate was 75.4%. Among 8150 participants in the 2016 KNHANES, 3655 participants younger than 40 years were excluded. In this survey, a pulmonary function test was performed by including participants aged 40 years or older. Among 4495 participants, 481 participants with COPD diagnosed by a physician through standardized interviews were excluded. Diagnosis of COPD was based on the forced expiratory volume in one second (FEV1)/forced vital capacity (FVC). As FEV1 and FVC were outcome variables of the current study, we excluded the participants with COPD in the analysis. Among 4014 participants, 602 participants who had missing values in the independent variables and 235 participants who had missing values in the outcome variables were excluded. In total, 3177 (weighted *n* = 19,770,902) participants were selected for analysis. Written informed consent was obtained from participants before completing the survey. This study was approved by the Institutional Review Board of Soonchunhyang University Bucheon Hospital (IRB number; SCHBC 2018-05-017, date of approval; 7 June 2018).

### 2.2. Demographic Variables and Data Collection

The KNHANES was conducted by four special research teams, each comprising of eight experts including nutritionists, nurses, and students with majors in public health. The selected professional investigator was placed at the investigation site after completion of education and practice for one month. Subsequently, the ability to conduct research was verified through regular education and an on-site quality management system. A standardized interview was performed in participants’ homes, and an established questionnaire was used to collect information about demographic variables and socioeconomic characteristics. Specifically, data on age, sex, income, region, education, marital status, alcohol consumption, and smoking status were collected from the participants. Alcohol consumption was divided into the following four groups based on the frequency of alcohol consumption during the past year: never, ≤1/week, 2 to 3/week, and ≥4/week. Income level was categorized into quartiles based on average individual monthly income. Urban and rural areas were classified by the administrative district [15]. Serum uric acid levels were measured by colorimetry with a uricase–catalase system (Hitachi automatic analyzer 7600–210, Tokyo, Japan). Estimating Glomerular Filtration Rate (eGFR) was calculated by using the Modification of Diet in Renal Disease (MDRD) study equation.
eGFR (mL/min/1.73 m^2^) = 175 × (Serum creatinine)^−1.154^ × (Age)^−0.203^ × (0.742 if female)(1)

Blood pressure was measured by nurses by using a standard protocol. Blood pressure was measured on the right arm using a mercury sphygmomanometer after at least five minutes of rest and a sitting position. Blood pressure was measured three times and the mean value of the second and third blood pressure was used for analysis. Mean arterial pressure (MAP) was calculated using the following formula:MAP = 1/3 × systolic blood pressure + 2/3 × diastolic blood pressure(2)

Blood samples were collected from participants after at least eight hours of fasting. The blood samples were immediately refrigerated and transported in cold storage to the central testing facility. Height and weight were assessed using standardized techniques and equipment. Briefly, height was measured to the nearest 0.1 cm using a portable stadiometer SECA 225 (SECA, Hamburg, Germany). Weight was measured to the nearest 0.1 kg using a GL-6000-20 (G-tech, Seoul, Korea). Body mass index (BMI) was calculated by dividing weight by height squared (kg/m^2^). Pulmonary tuberculosis, asthma, and lung cancer were defined in the questionnaire as ʻpulmonary tuberculosis diagnosed by a physicianʼ through a standardized interview. The question was “Was your pulmonary tuberculosis diagnosed by a physician?”. The questionnaire consisted of three responses (1, Yes; 2, No; 3, I have never been sick before). Participants who chose 1 (Yes) were classified under pulmonary tuberculosis. Each interview was conducted individually by a trained professional investigator. 

### 2.3. Measurement of Lung Function

Pulmonary function test was performed by Vyntus Spiro (CareFusion, San Diego, CA, USA). The subjects were required to be in a sitting posture with their upper body tilted about 15 degrees and asked to maintain the posture until the completion of the test. The test was demonstrated in easy-to-understand terms and the subjects were encouraged to exhale through exercises. Spirometry was conducted at least three times per subject (maximum up to eight times). Tests with acceptable curves and without errors were performed twice or more, and the best graph was stored, according to the American Thoracic Society/European Respiratory criteria for standardization [16]. Spirometric data was obtained on-site by clinical technicians and transferred to the central database. The FEV1, FVC values, and FEV1/FVC values were used for analysis. 

### 2.4. Statistical Analyses

To reflect representative estimates of the non-institutionalized Korean civilian population, survey sample weights were applied in the analyses. Sample weights were calculated by considering the sampling rate, response rate, and age/sex proportions of the reference population (2005 Korean National Census registry). Clinical characteristics were analyzed according to dichotomized serum UA or quartiles of UA. Due to the sex difference in UA levels, the definition of hyperuricemia was different in males and females (>7 mg/dL in males and >6 mg/dL in females) [10,17]. To analyze the effect of UA on pulmonary function test, univariable linear regression models were computed with FEV1, FVC, FEV1/FVC as dependent variables and hyperuricemia and UA quartiles as independent variables. Variables associated with *p*-value outcomes ≤0.05 in univariable analysis were included in the multivariable analysis. Multivariable linear regression analysis was performed as model 1 adjusting for age, income, region, education, marriage, smoking, alcohol consumption, BMI, eGFR, hemoglobin A1c (HbA1c), and MAP. Model 2 was obtained after adjusting model 1 plus pulmonary tuberculosis, asthma, and lung cancer. Simple correlation analysis was performed to evaluate the association between lung function and UA levels. Analyses were performed using SPSS statistics version 20.0 (IBM Corp., Armonk, NY, USA). All *p*-values were two-sided, and a *p*-value of less than 0.05 was considered to indicate statistical significance.

### 2.5. Patient and Public Involvement

The KNHANES data were released after anonymization and the study population was not involved in the design of this study.

## 3. Results

A total of 3177 (weighted *n* = 19,770,902) participants aged 40 years or older were selected from the 2016 KNHANES and analyzed. The mean age was 55.60 ± 0.27 years old. Among 3177 participants, 1401 (49.9%) were males and 1776 (50.1%) were females. Mean serum uric acid levels were 5.03 ± 0.02 mg/dL. The clinical characteristics of the subjects are summarized according to gender and the presence of hyperuricemia in Table 1. Female participants with hyperuricemia were older than participants with normouricemia. BMI, MAP, and HbA1c were significantly higher in the hyperuricemia group than the normouricemia group in both males and females. The eGFR was significantly lower in hyperuricemia group. FEV1 and FVC were significantly lower in hyperuricemia group in females. FVC was not significantly different between the two groups. However, in males, there was no significant difference in spirometric parameters according to the presence of hyperuricemia. 

Table 2 shows the clinical characteristics according to the UA quartile. An increase in UA quartile was significantly associated with higher BMI, MAP, and HbA1c, and was negatively associated with higher eGFR in both males and females. In males, UA quartile was positively associated with alcohol consumption. In females, UA quartile was positively associated with older age. An increased UA quartile was associated with decreased FEV1 and FVC in females. There was no significant association between UA quartile and lung function in males. 

Multivariable linear regression analysis was performed for effect of uric acid on pulmonary function (Table 3). Hyperuricemia was associated with decreased FEV1 and FVC in female participants after adjustment for age, income, region, education, marital status, alcohol consumption, smoking, BMI, MAP, HbA1c, and eGFR (model 1). After additional adjustment for respiratory disease, including pulmonary tuberculosis, asthma, and lung cancer, hyperuricemia was observed to be associated with decreased FEV1 and FVC (model 2). An increase in UA quartile was also significantly associated with decreased FEV1 and FVC in both model 1 and model 2 in females. However, there was no significant association between UA and lung function after adjustment (both model 1 and model 2) in male participants. 

Subgroup analysis was performed for participants with lung disease. Among participants with pulmonary tuberculosis, UA was not significantly associated with spirometric parameters after adjustment in both males and females. There was no significant association between UA and lung function in female participants with asthma. Multivariable linear regression analysis was not performed in male participants with asthma and in participants with lung cancer due to insufficient sample size.

## 4. Discussion

This study demonstrated that serum UA levels were inversely associated with lung function in the female general population aged 40 years or older. Increased serum UA levels were a significant predictor for decreased FEV1 and FVC independent of age, income, region, education, marital status, alcohol consumption, smoking, BMI, MAP, HbA1c, eGFR, and some pulmonary diseases in female participants. There was no significant association between UA and pulmonary function in male participants. 

Results of the current study are similar to that of the Takahata study, the community-based annual health check for general population aged 40 years or older in Japan, which reported that FEV1% predicted and FVC% predicted were inversely correlated with serum UA in female [9]. In males, although there was no significant relationship in univariable analysis, a significant inverse relationship was observed in multivariable analysis. The current study was inconsistent with a previous Korean study which reported that UA was positively associated with pulmonary function in the middle-aged healthy population [10]. The reported study excluded the subjects with a self-reported history of underlying diseases including chronic liver disease, cardiovascular disease, diabetes, chronic renal disease, malignancy, rheumatologic disease, arthritis, and chronic lung disease. Whereas, in the current study, we included patients with underlying diseases except for the participants with a self-reported history of COPD. The mean age of enrolled subjects was younger in the study by Song et al. [10] (40 years old) than our study (56 years old). We found that UA was positively associated with FEV1 and FVC in female general population regardless of the presence of underlying diseases.

An increase in the serum UA levels has been reported in hypoxia. UA/creatinine ratio is a marker of tissue hypoxia in patients with obstructive sleep apnea [18]. In patients with pulmonary thromboembolism, a significant elevation in serum UA has been reported followed by a decrease with treatment and is associated with an increase in cardiac output and arterial oxygen tension [19]. Serum UA levels were associated with severity of idiopathic pulmonary artery hypertension and right ventricular dysfunction [20]. Severe right ventricular failure associated with markedly reduced cardiac output ultimately induces hypoxia. Xanthine oxidase activity is regulated by hypoxia. Arterial endothelial cells exposed to hypoxia and anoxia demonstrated the enhancement in xanthine oxidase activity compared with controls [21]. Increased pulmonary artery pressures leads to increased right ventricle afterload, which results in purine degradation through increased xanthine oxidase activity [22]. Experimentally induced hypoxia models have shown that UA concentration is higher in hypoxic status compared to either normoxia or hypoxia in lung tissue, thus suggesting that pulmonary hypoxia results in greater purine catabolism leading to increased UA concentration [23].

Tumor lysis syndrome is characterized by an increased release of intracellular contents such as UA into the extracellular compartment [24]. A large amount of UA from injured tissue causes tissue damage and is associated with significant morbidity and mortality. Local high concentration of UA causes gout. The inflammatory response is initiated by monosodium urate crystal deposition in the joints, bones, and soft tissues, which activates NOD-like receptor protein 3 (NLRP3) inflammasome, which in turn activates caspase-1 and releases key cytokine interleukin (IL)-1β [25]. Gasse et al. [26] reported that lung injury depends on the NALP3 inflammasome. UA released from bleomycin injured cells triggers NALP3 inflammasome and IL-1β production, leading to lung injury and fibrosis. 

Hyperuricemia is associated with systemic inflammation. In a large population-based study, a positive significant association between serum UA and inflammatory markers such as C-reactive protein (CRP) and IL-6 has been observed [27]. UA stimulates proinflammatory response and upregulates cyclo-oxygenase-2 activity and vascular CRP in vascular smooth muscle cells [28]. UA also increases the production of IL-1β, IL-6, and tumor necrosis factor (TNF)-α in human mononuclear cells [29]. Inflammatory cytokines increase the UA production by increasing the xanthine oxidase activity. The levels of TNF-α and IL-1β were significantly correlated with the degree of xanthine oxidase activity in epithelial lining fluid of COPD patients [30]. 

In addition, elevated UA is associated with endothelial dysfunction [31]. Pulmonary endothelial dysfunction is associated with alveolar septal cell apoptosis and emphysema [32]. Pulmonary endothelial dysfunction associated with hyperuricemia may be linked to decreased pulmonary function [33]. 

Although the sample size was not sufficient to analyze the effect of UA on pulmonary function in subjects with all pulmonary disease in the present study, we found that there was no significant association between UA and lung function among participants with pulmonary tuberculosis in both males and females. Previous studies showed that hyperuricemia was associated with poor prognosis in patients with COPD. In the German COPD cohort study, UA was significantly associated with reduced FEV1, reduced 6 min walk distance, and acute exacerbations [33]. Hyperuricemia was associated with decreased FEV1 and a risk factor for mortality and acute exacerbation [8]. Furthermore, association with advanced duration and stage of COPD has been reported [34]. Serum UA levels were not significantly associated with lung function and symptoms in patients with asthma [35].

In the current study, there was no association between UA and pulmonary function in male participants. The cause of the sexual difference between pulmonary function and UA is not certain. In females, serum UA levels are lower than in males, but UA levels increase around menopause. Sex hormones may impact UA metabolism and the relative physiologic impact of hyperuricemia may be stronger among females than males [36]. 

This study has several limitations. First, although we analyzed pulmonary disease including asthma, pulmonary tuberculosis, and lung cancer, other pulmonary diseases such as interstitial lung disease and bronchiectasis were not identified in this survey. Second, it was not possible to analyze medications such as UA lowering agents or diuretics that could affect the serum UA levels. Third, biochemical characteristics that closely related serum uric acid levels were included in the analysis. Inflammatory biochemical parameters which could affect pulmonary function were not included in this study. Fourth, the causality could not be determined due to the cross-sectional design of this study. Despite all these limitations, the strength of this study lies in the fact that we analyzed the relationship between UA and pulmonary function using a nationwide representative sample of the general adult population. In conclusion, serum UA was negatively associated with FEV1 and FVC in the Korean female general population. Results of this study suggest that hyperuricemia may decrease pulmonary function. Further investigations are required to clarify the relationship between serum UA levels and pulmonary function.

## Figures and Tables

**Table 1 medicina-57-00953-t001:** Clinical and demographic characteristics of participants with hyperuricemia and with normal uric acid levels.

Variables	Male (Unweighted *n* = 1401)			Female (Unweighted *n* = 1776)		
	Normal UA UA < 7 mg/dL (*n* = 1161)	Hyperuricemia UA ≥ 7 mg/dL (*n* = 240)	*p* Value	Normal UA UA < 6 mg/dL (*n* = 1648)	Hyperuricemia UA ≥ 6 mg/dL (*n* = 128)	*p* Value
Age, mean (years)	55.00 ± 0.31	53.86 ± 0.68	0.067	55.83 ± 0.34	61.30 ± 1.26	<0.001
Income			0.049			0.472
Low	255 (22.0)	70 (29.2)		377 (22.9)	32 (25.0)	
Mid-low	298 (25.7)	53 (22.1)		416 (25.2)	38 (29.7)	
Mid-high	296 (25.5)	61 (25.4)		415 (25.2)	33 (25.8)	
High	312 (26.9)	56 (23.3)		440 (26.7)	25 (19.5)	
Education			0.287			0.072
Elementary school	224 (19.3)	35 (14.6)		511 (31.0)	54 (42.2)	
Middle school	158 (13.6)	34 (14.2)		242 (14.7)	13 (10.2)	
High school	356 (30.7)	80 (33.3)		505 (30.6)	43 (33.6)	
College graduation	423 (36.4)	91 (37.9)		390 (23.7)	18 (14.1)	
Region			0.780			0.541
Urban	909 (78.3)	188 (78.3)		1315 (79.8)	103 (80.5)	
Rural	252 (21.7)	52 (21.7)		333 (20.2)	25 (19.5)	
Marital status			0.049			-
Married	1118 (96.3)	224 (93.3)		1620 (98.3)	128 (100.0)	
Unmarried	41 (3.5)	16 (6.7)		28 (1.7)	0 (0.0)	
Alcohol consumption			0.068			0.428
Never	235 (20.2)	44 (18.3)		650 (39.4)	61 (47.7)	
≤1/week	497 (42.8)	84 (35.0)		842 (51.1)	57 (44.5)	
2–3/week	269 (23.2)	58 (24.2)		121 (7.3)	8 (6.3)	
≥4/week	160 (13.8)	54 (22.5)		35 (2.1)	2 (1.6)	
Smoking			0.914			0.569
Never smoker	212 (18.3)	35 (14.6)		1523 (92.4)	116 (90.6)	
Ex-smoker	590 (50.8)	123 (51.3)		50 (3.0)	4 (3.1)	
Current smoker	359 (30.9)	82 (34.2)		75 (4.6)	8 (6.3)	
Pulmonary tuberculosis	63 (5.4)	16 (6.7)	0.655	64 (3.9)	3 (2.3)	0.981
Asthma	13 (1.1)	6 (2.5)	0.274	59 (3.6)	5 (3.9)	0.244
Lung cancer	5 (0.4)	2 (0.8)	0.459	2 (0.1)	0 (0.0)	-
BMI (kg/m^2^)	24.54 ± 0.11	25.47 ± 0.21	<0.001	24.03 ± 0.11	25.99 ± 0.42	<0.001
eGFR	85.98 ± 0.99	76.89 ± 1.01	<0.001	88.65 ± 0.53	73.28 ± 2.13	<0.001
MAP (mmHg)	93.79 ± 0.38	95.48 ± 0.79	0.052	89.94 ± 0.36	92.37 ± 1.05	0.024
Hemoglobin A1c (%)	5.87 ± 0.03	5.66 ± 0.03	<0.001	5.73 ± 0.02	6.00 ± 0.07	<0.001
FEV1 (L)	3.17 ± 0.02	3.23 ± 0.04	0.218	2.36 ± 0.02	2.08 ± 0.05	<0.001
FVC (L)	4.17 ± 0.02	4.22 ± 0.06	0.420	2.96 ± 0.02	2.65 ± 0.06	<0.001
FEV1/FVC	0.76 ± 0.00	0.76 ± 0.00	0.203	0.80 ± 0.00	0.78 ± 0.01	0.116

Values are presented as mean ± standard deviation or *n* (weighted %). BMI: body mass index, eGFR: estimated glomerular filtration rate, MAP: mean arterial pressure, FEV1: forced expiratory volume in 1 second, FVC: forced vital capacity, UA: uric acid.

**Table 2 medicina-57-00953-t002:** Clinical and demographic characteristics according to serum uric acid quartiles.

	Male					Female				
	Q1 * (*n* = 357)	Q2 ^†^ (*n* = 379)	Q3 ^‡^ (*n* = 323)	Q4 ^§^ (*n* = 342)	*p* Value	Q1 ^∥^ (*n* = 470)	Q2 ^¶^ (*n* = 445)	Q3 ** (*n* = 409)	Q4 ^††^ (*n* = 452)	*p* Value
Age, mean (years)	56.20 ± 0.55	55.50 ± 0.54	54.51 ± 0.57	53.68 ± 0.59	0.016	54.81 ± 0.53	55.18 ± 0.60	55.97 ± 0.53	59.02 ± 0.70	<0.001
Income					0.209					0.241
Low	73 (20.4)	93 (24.5)	67 (20.7)	92 (26.9)		100 (21.3)	99 (22.2)	112 (27.4)	98 (21.7)	
Mid-low	110 (30.8)	88 (23.2)	76 (23.5)	77 (22.5)		129 (27.4)	118 (26.5)	97 (23.7)	110 (24.3)	
Mid-high	85 (23.8)	94 (24.8)	93 (28.8)	85 (24.9)		122 (26.0)	116 (26.1)	87 (21.3)	123 (27.2)	
High	89 (24.9)	104 (27.4)	87 (26.9)	88 (25.7)		119 (25.3)	112 (25.2)	113 (27.6)	121 (26.8)	
Education					0.005					0.723
Elementary school	73 (20.4)	76 (20.1)	54 (16.7)	56 (16.4)		138 (29.4)	156 (35.1)	113 (27.6)	158 (35.0)	
Middle school	56 (15.7)	50 (13.2)	36 (11.1)	50 (14.6)		74 (15.7)	49 (11.0)	69 (16.9)	63 (13.9)	
High school	127 (35.6)	113 (29.8)	89 (27.6)	107 (31.3)		148 (31.5)	138 (31.0)	125 (30.6)	137 (30.3)	
College graduation	101 (28.3)	140 (36.9)	144 (44.6)	129 (37.7)		110 (23.4)	102 (22.9)	102 (24.9)	94 (20.8)	
Region					0.075					0.044
Urban	271 (75.9)	296 (78.1)	267 (82.7)	263 (76.9)		380 (80.9)	366 (82.2)	328 (80.2)	344 (76.1)	
Rural	86 (24.1)	83 (21.9)	56 (17.3)	79 (23.1)		90 (19.1)	79 (17.8)	81 (19.8)	108 (23.9)	
Marital status					0.181					0.075
Married	345 (96.6)	365 (96.3)	312 (96.6)	320 (93.6)		468 (99.6)	438 (98.4)	399 (97.6)	443 (98.0)	
Unmarried	12 (3.4)	14 (3.7)	11 (3.4)	22 (6.4)		2 (0.4)	7 (1.6)	10 (2.4)	9 (2.0)	
Alcohol consumption					0.005					0.063
Never	82 (23.0)	93 (24.5)	50 (15.5)	54 (15.8)		190 (40.4)	168 (37.8)	151 (36.9)	202 (44.7)	
≤1/week	131 (36.7)	173 (45.6)	149 (46.1)	128 (37.4)		243 (51.7)	239 (53.7)	207 (50.6)	210 (46.5)	
2-3/week	84 (23.5)	77 (20.3)	77 (23.8)	89 (26.0)		32 (6.8)	29 (6.5)	41 (10.0)	27 (6.0)	
≥4/week	60 (16.8)	36 (9.5)	47 (14.6)	71 (20.8)		5 (1.1)	9 (2.0)	10 (2.4)	13 (2.9)	
Smoking					0.139					0.036
Never smoker	60 (16.8)	68 (17.9)	64 (19.8)	55 (16.1)		441 (93.8)	418 (93.9)	371 (90.7)	409 (90.5)	
Ex-smoker	168 (47.1)	208 (54.9)	162 (50.2)	175 (51.2)		13 (2.8)	9 (2.0)	13 (3.2)	19 (4.2)	
Current smoker	129 (36.1)	103 (27.2)	97 (30.0)	112 (32.7)		16 (3.4)	18 (4.0)	25 (6.1)	24 (5.3)	
Pulmonary tuberculosis	24 (6.7)	19 (5.0)	14 (4.3)	22 (6.4)	0.142	18 (3.8)	26 (5.8)	12 (2.9)	11 (2.4)	0.075
Asthma	4 (1.1)	1 (0.3)	5 (1.5)	9 (2.6)	0.003	9 (1.9)	12 (2.7)	19 (4.6)	24 (5.3)	0.014
Lung cancer	2 (0.6)	1 (0.3)	2 (0.6)	2 (0.6)	0.608	1 (0.2)	0 (0.0)	1 (0.2)	0 (0.0)	-
BMI (kg/m^2^)	24.31 ± 0.20	24.35 ± 0.17	24.88 ± 0.18	25.31 ± 0.18	<0.001	23.25 ± 0.16	23.81 ± 0.17	24.42 ± 0.21	25.30 ± 0.23	<0.001
eGFR	91.32 ± 2.80	84.53 ± 0.83	83.16 ± 0.80	78.23 ± 0.86	<0.001	93.34 ± 0.92	89.50 ± 0.81	86.90 ± 0.90	79.91 ± 0.99	<0.001
MAP (mmHg)	93.67 ± 0.63	94.08 ± 0.58	93.69 ± 0.71	94.87 ± 0.69	0.518	88.86 ± 0.55	89.41 ± 0.68	90.74 ± 0.65	91.64 ± 0.54	<0.001
Hemoglobin A1c (%)	6.09 ± 0.08	5.84 ± 0.04	5.69 ± 0.04	5.69 ± 0.04	<0.001	5.74 ± 0.05	5.68 ± 0.04	5.73 ± 0.03	5.86 ± 0.04	0.003
FEV1 (L)	3.10 ± 0.04	3.17 ± 0.04	3.19 ± 0.04	3.26 ± 0.04	0.051	2.37 ± 0.03	2.41 ± 0.02	2.34 ± 0.03	2.23 ± 0.03	<0.001
FVC (L)	4.13 ± 0.04	4.15 ± 0.04	4.17 ± 0.04	4.27 ± 0.05	0.124	2.99 ± 0.04	3.00 ± 0.03	2.95 ± 0.03	2.80 ± 0.03	<0.001
FEV1/FVC	0.75 ± 0.00	0.76 ± 0.00	0.76 ± 0.00	0.76 ± 0.00	0.115	0.79 ± 0.00	0.80 ± 0.00	0.79 ± 0.00	0.79 ± 0.00	0.133

BMI: body mass index, eGFR: estimated glomerular filtration rate, MAP: mean arterial pressure, FEV1: forced expiratory volume in 1 second, FVC: forced vital capacity, Q: quartile. * <4.8 mg/dL. ^†^ 4.8–5.7 mg/dL. ^‡^ 5.7–6.5 mg/dL. ^§^ >6.5 mg/dL. ^∥^ <3.7 mg/dL. ^¶^ 3.7–4.3 mg/dL. ** 4.3–4.9 mg/dL. ^††^ >4.9 mg/dL.

**Table 3 medicina-57-00953-t003:** Multivariable linear regression analysis for effect of uric acid on pulmonary function. For comparison with lowest uric acid quartile.

	Hyperuricemia		Uric Acid Quartile				
	β ± SE	*p* Value	Q1	Q2 β ± SE	Q3 β ± SE	Q4 β ± SE	*p* Value
Model 1 ^†^							
Male							
FEV1	−0.008 ± 0.038	0.841	Ref.	0.013 ± 0.039	−0.028 ± 0.046	0.022 ± 0.044	0.687
FVC	−0.016 ± 0.050	0.753	Ref.	−0.030 ± 0.050	−0.070 ± 0.055	0.019 ± 0.055	0.374
FEV1/FVC	0.001 ± 0.005	0.841	Ref.	0.009 ± 0.006	0.006 ± 0.006	0.002 ± 0.006	0.428
Female							
FEV1	−0.143 ± 0.046	0.002	Ref.	0.033 ± 0.027	−0.021 ± 0.034	−0.053 ± 0.031	0.006
FVC	−0.159 ± 0.048	0.001	Ref.	0.015 ± 0.034	−0.024 ± 0.041	−0.079 ± 0.041	0.021
FEV1/FVC	−0.008 ± 0.008	0.308	Ref.	0.006 ± 0.004	0.000 ± 0.005	0.001 ± 0.004	0.351
Model 2 ^‡^							
Male							
FEV1	0.000 ± 0.038	0.996	Ref.	0.004 ± 0.038	−0.029 ± 0.046	0.028 ± 0.042	0.631
FVC	−0.011 ± 0.051	0.832	Ref.	−0.039 ± 0.049	−0.076 ± 0.055	0.017 ± 0.054	0.314
FEV1/FVC	0.002 ± 0.005	0.666	Ref.	0.008 ± 0.006	0.007 ± 0.006	0.004 ± 0.006	0.545
Female							
FEV1	−0.142 ± 0.044	0.001	Ref.	0.039 ± 0.026	−0.018 ± 0.034	−0.050 ± 0.031	0.002
FVC	−0.161 ± 0.047	<0.001	Ref.	0.020 ± 0.034	−0.026 ± 0.040	−0.083 ± 0.040 *	0.007
FEV1/FVC	−0.008 ± 0.008	0.307	Ref.	0.006 ± 0.003	0.001 ± 0.004	0.003 ± 0.004	0.255

FEV1: forced expiratory volume in 1 second, FVC: forced vital capacity, β: unstandardized regression coefficient, SE: standard error, Q: quartile, Ref.: reference. * *p* < 0.05. ^†^ Model 1: adjusted for age, income, region, education, marital status, alcohol consumption, smoking, body mass index, mean arterial pressure, hemoglobin A1c, and estimated glomerular filtration rate. ^‡^ Model 2: adjusted for model 1 plus pulmonary tuberculosis, asthma, and lung cancer.

## Data Availability

Data and material are available on reasonable request.

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
