# Peer review of "Gender-Specific Association of Serum Uric Acid and Pulmonary Function: Data from the Korea National Health and Nutrition Examination Survey"

_medicina, 2021, doi:10.3390/medicina57090953_

Round 1

Reviewer 1 Report

In the present manuscript by Hyemin Jeong et al. titled »The Relationship between Serum Uric Acid and Pulmonary 2 Function: Data from the Korea National Health and Nutrition 3 Examination Survey« the authors aim to evaluate the relationship between serum UA and pulmonary function 70 in the Korean adult population using nationwide data. The manuscript represents a cross-sectional analysis of the survey results of participants older than 40 years without COPD that participated The Korea National Health and Nutrition Examination Survey (KNHANES) from the year 2016.

The manuscript is very well written and provides further insight and strengthening of the information that serum uric acid could prove as a valuable biomarker for pulmonary disease, although currently cannot be causally associated to pulmonary disease. I believe that the manuscript could be accepted with minor revision. I have provided some suggestions on ways to improve the manuscript, which are provided in the comments below.

Comments regarding minor revision:

  • Serum uric acid is mostly excreted through the kidneys, which is why kidney function is important as the authors have stated. An estimate of the kidney function was used in the analysis, however, it is not clear what serum biomarker (probably creatinine) and which estimation equation was used (if creatinine was used I suggest they use CKD-EPI which is the current standard).
  • In Table 1 and 2 several variables are presented with units. For MAP mm per Hg (mm/Hg) is provided, which should be mmHg.
  • The results section could be further enhanced for comprehensiveness with a short introduction of the cohort (which is already performed in the methods section but could be transferred here. Information on number of participants, gender distribution, whole cohort mean+SD and range of UA could be provided, etc.
  • The authors have cited 36 manuscripts in the references, however only 11% (4/36) were published in the last 5 years and 25% (9/36) were published in the last 10 years. I suggest that the reference list could be updated.

Author Response

Thank you for your insight and comments on our manuscript. We have revised our manuscript in response to the reviewers’ comments. With these changes, we believe that the manuscript has been greatly improved. Our point-by-point responses to the reviewers’ comments are described below.

Reviewer 1.

C1. Serum uric acid is mostly excreted through the kidneys, which is why kidney function is important as the authors have stated. An estimate of the kidney function was used in the analysis, however, it is not clear what serum biomarker (probably creatinine) and which estimation equation was used (if creatinine was used I suggest they use CKD-EPI which is the current standard).

R1. Thank you for your comment. We used serum creatinine to estimate kidney function. In current study, the Modification of Diet in Renal Disease (MDRD) study equation was used for estimating GFR.

GFR (mL/min/1.73 m2) = 175 × (Scr)-1.154 × (Age)-0.203 × (0.742 if female).

We have added this information in the method section of revised manuscript (lines 117-119). In the future, we will consider CKD-EPI equation in another study.  

C2. In Table 1 and 2 several variables are presented with units. For MAP mm per Hg (mm/Hg) is provided, which should be mmHg.

R2. Thank you for your comment. We made a mistake. We have corrected it in the revised manuscript.

C3. The results section could be further enhanced for comprehensiveness with a short introduction of the cohort (which is already performed in the methods section but could be transferred here. Information on number of participants, gender distribution, whole cohort mean+SD and range of UA could be provided, etc.

R3. Thank you for your comment. We have added short introduction of the cohort in the results section (Lines 173-176).

A total of 3,177 (weighted n = 19,770,902) participants aged 40 years or older were selected from the 2016 KNHANES and analyzed. Mean age was 55.60 ± 0.27 years old. Among 3,177 participants, 1,401 (49.9%) were males and 1,776 (50.1%) were females. Mean serum uric acid levels were 5.03 ± 0.02 mg/dL

C4. The authors have cited 36 manuscripts in the references, however only 11% (4/36) were published in the last 5 years and 25% (9/36) were published in the last 10 years. I suggest that the reference list could be updated.

R4. We have updated the reference list. Among 36 references, 38.9% (14/36) were published within last 5 years and 55.6% (20/36) were published within the last 10 years.

Thank you for the opportunity to revise and resubmit our manuscript. We hope that the revised manuscript now is considered acceptable for publication in medicina.

Reviewer 2 Report

1) The Authors performed a descriptive analysis of  parameters according to sex but this is not evident from the title and the abstract. If this is a main result please change title and abstract accordingly. 

2) Authors should indicate the uric acid dosage method.

3) In the present study there are very few biochemical characteristic of the study population, authors reported analyzed only hemoglobin A1c and e-GFR. This is a criticism. Moreover authors should indicate why they did not investigated inflammatory profile of the study population. 

4) In the discussion authors conclude  that there was no association between UA and pulmonary function in male participants, but the cause of the sexual difference between pulmonary function and UA is not certain. In females, serum UA levels was lower than males, but UA levels increased 
around menopause. Female population with hyperuricemia is older than female population with normal UA (p<0.0001), while there are not statistically significant differences between male population with hyperuricemia and male population with normal UA (p=0.067). What about the comparison between female population vs male population? Are they matched for age? 

4) Table  1 and 2, educational, marital status, region are not clinical characteristics and comorbidities, but demographic characteristics. Please correct the titles of the tables. 

5) Tabe 3 is not clear, please explain it in detail.

6) Line 153 please provide a citation with UA guidelines. 

Author Response

Thank you for your insight and comments on our manuscript. We have revised our manuscript in response to the reviewers’ comments. With these changes, we believe that the manuscript has been greatly improved. Our point-by-point responses to the reviewers’ comments are described below.

Reviewer 2.

C1. The Authors performed a descriptive analysis of parameters according to sex but this is not evident from the title and the abstract. If this is a main result please change title and abstract accordingly. 

R1. Thank you for your comments. We have changed the title and the abstract in the revised manuscript. Title was changed as `Gender-specific association of serum uric acid and pulmonary function: Data from the Korean National Health and Nutrition Examination Survey (Lines 2-4, 26-27).

C2. Authors should indicate the uric acid dosage method.

R2. We have added the information about measurement of uric acid in the method section. Serum uric acid levels were measured by colorimetry with a uricase–catalase system (Hitachi automatic analyzer 7600–210, Tokyo, Japan), (Lines 115-117).

C3. In the present study there are very few biochemical characteristic of the study population, authors reported analyzed only hemoglobin A1c and e-GFR. This is a criticism. Moreover authors should indicate why they did not investigated inflammatory profile of the study population. 

R3. Thank you for your comment. We included biochemical factors that closely related serum uric acid levels. Inflammatory biochemical parameters which could affect pulmonary function were not included in this study. This is limitation of our study. We have added this limitation in the discussion session (Lines 303-305).

C4. In the discussion authors conclude that there was no association between UA and pulmonary function in male participants, but the cause of the sexual difference between pulmonary function and UA is not certain. In females, serum UA levels was lower than males, but UA levels increased around menopause. Female population with hyperuricemia is older than female population with normal UA (p<0.0001), while there are not statistically significant differences between male population with hyperuricemia and male population with normal UA (p=0.067). What about the comparison between female population vs male population? Are they matched for age? 

R4. The KNHANES is a nationwide survey conducted periodically to investigate health and nutritional status of the Korean population. Participants were selected using the proportional allocation-systematic sampling method with multistage stratification to derive a representative Korean population. Age is included in the stratification. In the KNHANES, a pulmonary function test was performed for participants aged 40 years or older. In this study, a total of 3,177 (weighted n = 19,770,902) participants aged 40 years or older were included from the KNHANES. Among 3,177 participants, 1,401 (49.9%) were males and 1,776 (50.1%) were females. Mean age of male participants was 55.0 ± 0.3 years old. Mean age of female participants was 56.1 ± 0.3 years old. Age was matched between male vs female population.

C5. Table 1 and 2, educational, marital status, region are not clinical characteristics and comorbidities, but demographic characteristics. Please correct the titles of the tables. 

R5. Thank you for your comments. We have corrected the titles of the table 1 and table 2 (Lines 185 and 196).

C6. Table 3 is not clear, please explain it in detail.

R6. We have added the explanation about table 3 in the revised manuscript.

Multivariable linear regression analysis was performed for effect of uric acid on pulmonary function (Table 3). Hyperuricemia was associated with decreased FEV1 and FVC in female participants after adjustment for age, income, region, education, marital status, alcohol consumption, smoking, BMI, MAP, HbA1c, and eGFR (model 1). After additional adjustment for respiratory disease, including pulmonary tuberculosis, asthma, and lung cancer, hyperuricemia was observed to be associated with decreased FEV1 and FVC (model 2). An increase in UA quartile was also significantly associated with decreased FEV1 and FVC in both model 1 and model 2 in females. However, there was no significant association between UA and lung function after adjustment (both model 1 and model 2) in male participants (Lines 200-209).

C7. Line 153 please provide a citation with UA guidelines. 

R7. Hyperuricemia was defined as > 7.0 mg/dL of serum uric acid in men and ≥ 6.0 mg/dL in women as the standard definition for most studies. We have added the references in the revised manuscript (Line 157).

Thank you for the opportunity to revise and resubmit our manuscript. We hope that the revised manuscript now is considered acceptable for publication in Medicina.

Round 2

Reviewer 2 Report

Agree for pubblication